# Gate-tunable superconducting diode effect in a three-terminal Josephson device

Mohit Gupta [1], Gino V. Graziano[1], Mihir Pendharkar [2,3], Jason T. Dong [4], Connor P. Dempsey [2], Chris Palmstrøm [2,4,5] & Vlad S. Pribiag [1] ✉

The phenomenon of non-reciprocal critical current in a Josephson device, termed the Josephson diode effect, has garnered much recent interest. Realization of the diode effect requires inversion symmetry breaking, typically obtained by spin-orbit interactions. Here we report observation of the Josephson diode effect in a three-terminal Josephson device based upon an InAs quantum well two-dimensional electron gas proximitized by an epitaxial aluminum superconducting layer. We demonstrate that the diode efficiency in our devices can be tuned by a small out-of-plane magnetic field or by electrostatic gating. We show that the Josephson diode effect in these devices is a consequence of the artificial realization of a current-phase relation that contains higher harmonics. We also show nonlinear DC intermodulation and simultaneous two-signal rectification, enabled by the multi-terminal nature of the devices. Furthermore, we show that the diode effect is an inherent property of multi-terminal Josephson devices, establishing an immediately scalable approach by which potential applications of the Josephson diode effect can be realized, agnostic to the underlying material platform. These Josephson devices may also serve as gate-tunable building blocks in designing topologically protected qubits.

In conventional semiconducting diodes, the value of resistance depends upon the direction of the current flow. This asymmetry has been exploited in applications such as photodetectors, signal rectifiers, and oscillators[1,2]. A superconducting diode would have dissipationless supercurrent flowing upon application of one current bias polarity, while applying an equivalent bias in the reverse direction would produce conventional dissipative current. Josephson junctions (JJs) with non-reciprocal critical current, i.e. the magnitude of the critical current is dependent upon bias direction, are one avenue to realize a superconducting diode. Recent experimental observations[3–13] of this effect rely on inherent properties of the materials used to fabricate the devices, with proposed physical mechanisms including breaking of inversion and time reversal symmetries[3,14–17] and exotic superconductivity[18]. Asymmetric superconducting quantum

interference devices (SQUIDs) can also realize non-reciprocal transport[19–25] by using large and imbalanced self-inductances, typically in all-metallic Josephson devices, which has precluded electrostatic gating. In a recent work, it was also theorized that interferometers based upon higher-harmonic Josephson junctions can realize the Josephson diode effect, with implementation relying on highly transparent quantum point contact JJs[26]. These approaches are material specific and difficult to scale for potential applications of the Josephson diode effect, such as dissipationless electronics.

Experimental investigations of Andreev bound state spectra in multi-terminal Josephson devices have attracted considerable attention recently due to proposed topologically protected subgap states[27–33]. Despite technical challenges in realizing these subgap states, other interesting transport phenomena such as multi-terminal

[1]School of Physics and Astronomy, University of Minnesota, Minneapolis, MN 55455, USA. [2]Electrical and Computer Engineering, University of California Santa Barbara, Santa Barbara, CA 93106, USA. [3]Materials Science and Engineering, Stanford University, Stanford, CA 94305, USA. [4]Materials Department, University of California Santa Barbara, Santa Barbara, CA 93106, USA. [5]California NanoSystems Institute, University of California Santa Barbara, Santa Barbara, CA 93106, USA. ✉e-mail: vpribiag@umn.edu

Andreev reflections[34–36], fractional Shapiro steps[37,38], correlated phase dynamics[39,40], and semiclassical topological states[41] have been demonstrated. The discovery and characterization of non-reciprocal supercurrent flow in these devices stands to impact and expand upon these phenomena.

In this work, we show experimentally that the Josephson diode effect can occur in a relatively simple platform: a three-terminal Josephson device based on an InAs two-dimensional electron gas (2DEG) proximitized by an epitaxial aluminum layer[34,35]. We show by means of simulations that this diode effect is a consequence of the synthetic realization of a Josephson current-phase relation (CφR) that contains higher harmonic terms with a phase difference between them provided by a finite applied magnetic field. We further show that this diode can be switched between positive polarity, i.e. the positive-bias critical current ($I_c^+$) is larger than the negative-bias critical current ($|I_c^-|$), and negative polarity ($|I_c^-|>I_c^+$) by means of a small out-of-plane magnetic field or electrostatic gating. This also establishes, more generally, that the Josephson diode effect can be realized in any material system exhibiting the conventional CφR ($I \propto \sin\varphi$). Moreover, these devices may also serve as gate tunable building blocks of superconducting circuits to realize topologically protected qubits[42–46].

## Results

### Three-terminal Josephson diode

The three-terminal devices presented in this work are fabricated on InAs 2DEG proximitized by a superconducting epitaxial aluminum layer. The heterostructure used is shown in Fig. 1b. The Al layer is etched in a Y-shape in the central area of the mesa. The legs of this Y-shaped region form the three-terminal Josephson device. The three superconducting terminals of the device are labeled 1, 2 and 0. The spacing between the superconducting electrodes is measured by SEM to be $W \approx 150$ nm. Three independently tunable Ti/Au top gates cover each leg of the Y-shaped region, leaving a small ungated region in the center. This allows for independent control of the coupling between terminals. A false-colored scanning electron micrograph of the device is shown in Fig. 1a.

When no gate voltage is applied, the contribution to charge transport via the small common central region can be neglected compared to the dominant transport via the legs of the Y-shaped junction, as illustrated in Fig. 1c. A simplified model for the device can then be a circuit with three Josephson junctions connected in a triangular configuration, as shown in Fig. 2a. For source terminal 1 and drain terminal 0, the current conservation leads to the relations:

$$I_1 = I_c^{01}\sin(\varphi_1) + I_c^{12}\sin(\varphi_1 - \varphi_2 + \phi_e) \tag{1}$$

If the current through terminal 2 is zero, then

$$\varphi_2 = \varphi_1 - \varphi_2 + \phi_e + 2n\pi \tag{2}$$

where $I_c^{ij}$ is the critical current between terminals $i$ and $j$. The phase differences $\varphi_1$ and $\varphi_2$ are between terminals 1 and 0 and terminals 2 and 0 respectively. The additional phase $\phi_e$ is introduced by an applied magnetic flux. If the three JJs are identical, then the critical currents in all three arms are equal, $I_c^{01} = I_c^{02} = I_c^{12} = I_c$, resulting in the effective CφR:

$$I_1 = I_c\sin(\varphi_2) + I_c\sin(2\varphi_2 - \phi_e - 2n\pi) \tag{3}$$

or, in terms of $\phi_1$:

$$I_1 = I_c\sin(\varphi_1) + I_c\sin\left(\frac{\varphi_1 + \phi_e - 2n\pi}{2}\right) \tag{4}$$

The difference between the positive-bias critical current ($I_c^+$) and negative-bias critical current ($I_c^-$), $\delta I_c = I_c^+ - |I_c^-|$, is calculated numerically by taking the difference between the maximum and minimum values of Eq. (3) or (4). This is plotted as the red line in Fig. 2b, showing that the diode polarity is $\pi$-periodic. If instead terminal 2 is the source and terminal 0 the drain, this calculation gives the opposite polarity of $\delta I_c$ compared to the previous case, shown by dashed blue line in Fig. 2b.

Note that in the case of unequal critical currents between the three arms, Eq. (2) may contain additional terms with different amplitudes. However, the diode effect is still expected in that case[26]. These results can be extended to devices with more than three terminals, establishing multi-terminal Josephson devices as a generic platform to realize the Josephson diode effect. It is important to note that the origin of the diode effect in this system is related to the breaking of inversion symmetry by the device configuration. The choice of source and drain terminals confers a chirality to the circuit which breaks inversion symmetry, giving rise to the Josephson diode effect. These results are independent of the material platform used. As shown in our derivation, the third terminal does not need to be connected electrically to the outside world to realize the diode effect, but its presence is required to obtain the necessary current-phase relation.

### Non-reciprocal Supercurrent Flow

Current-biased DC measurements are performed using configuration shown in Fig. 1a in a Triton dilution refrigerator at a base temperature of 14 mK. Terminals 1 and 0 are biased by a DC current source, labeled

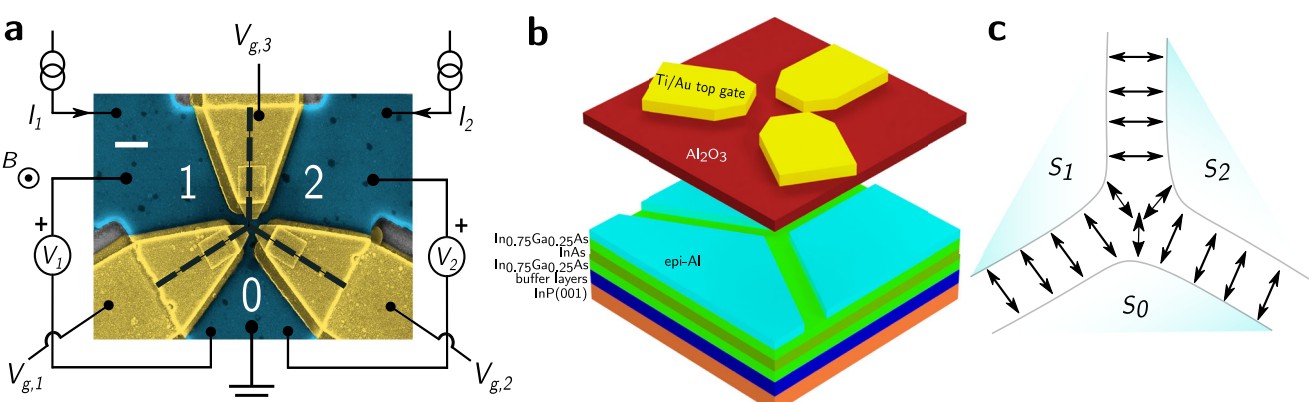

**Fig. 1 | Device architecture. a** False-color scanning electron microscope (SEM) image of Device 1, a three-terminal Josephson device with individually tunable gates, showing measurement schematic. The etched junction area is shown by a dashed black line between the Al layer (blue-colored) under the gates (gold-colored). The scale bar is 1 $\mu$m. **b** 3D schematic of the device showing layered heterostructure. **c** Schematic of transport in the three-terminal Josephson device, with arrows indicating current flow.

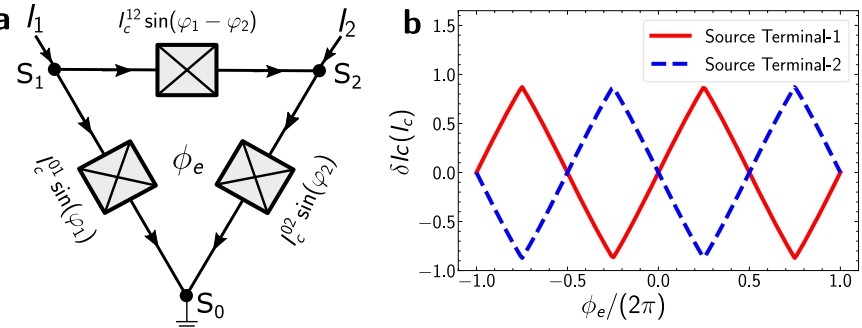

**Fig. 2 | Idealized network model. a** Effective transport model based upon three JJs exhibiting conventional C$\varphi$Rs connected in a triangular network, with arrows showing directions of the current flow for the configuration in which terminal 1 is the source and terminal 0 the drain. Flux threaded by the external field in the central common region is denoted by $\phi_e$. **b** $\delta I_c$ obtained from the configuration shown in panel **a**; $\delta I_c$ for source terminal 1 is shown by the red solid line, and for source terminal 2 by the dashed blue line.

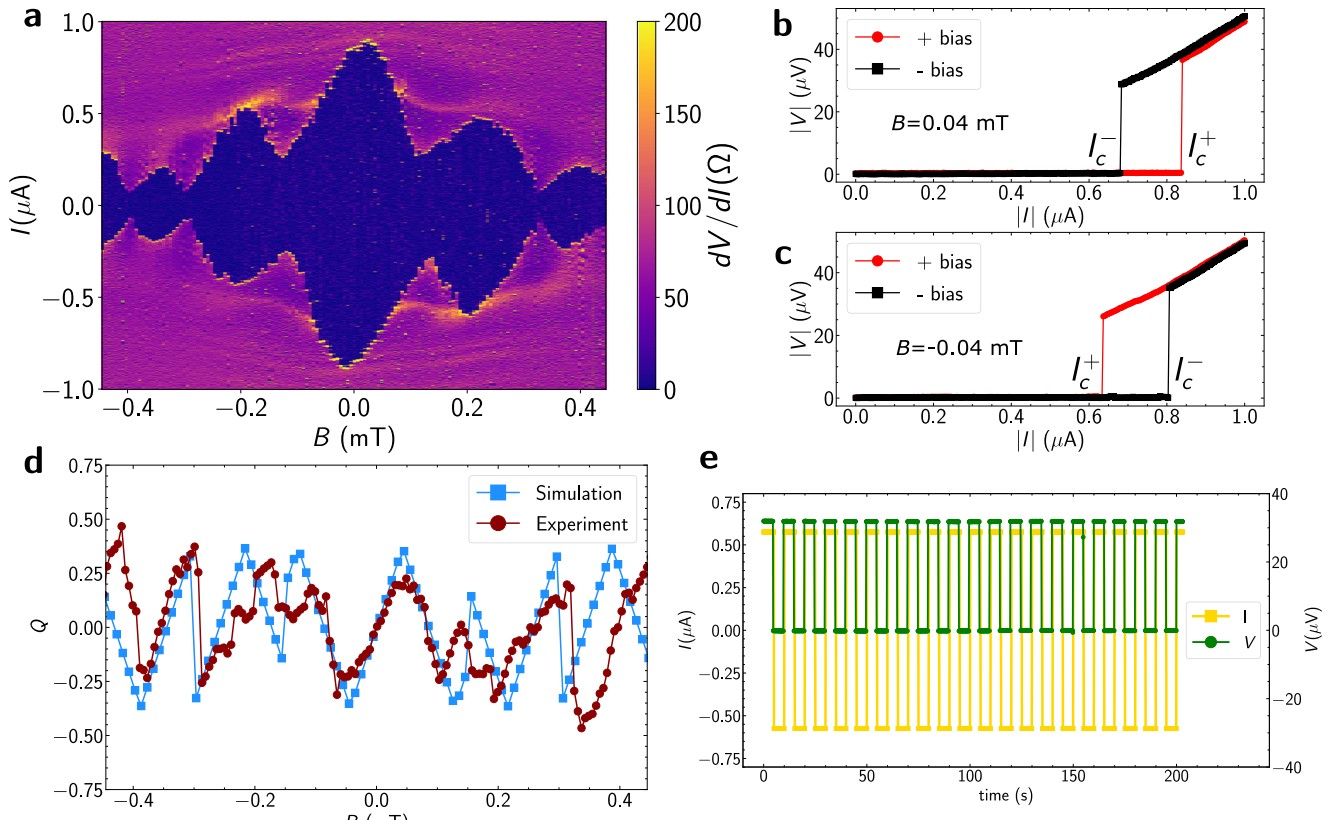

**Fig. 3 | Josephson diode effect. a** Differential resistance map of Device 1 as a function of bias current and applied out-of-plane magnetic field, $B$. **b** $I$–$V$ characteristics for both bias directions at $B = 0.04$ mT. **c** Identical measurement as **b**, but at $B = -0.04$ mT. The current sweeps always start at $I = 0$. **d** Diode efficiency factor $Q$ as a function of $B$ obtained from experiment (red circles) and simulations (blue squares). **e** Measured voltage drop (shown in green) for an applied square wave current pulse (shown in yellow).

$I_1 = I$, and the voltage across the same terminals, $V_1 = V$ is measured. It is important to note that in our measurements terminal 2 is floating unless stated otherwise. In practice terminal 2 can be a superconducting island with no possibility of galvanic connection and hence the device can function as a Josephson diode with only two active terminals with an effective C$\varphi$R. An external out-of-plane magnetic field, $B$, is applied using a superconducting magnet. To measure the critical current, the differential resistance d$V$/d$I$ is calculated as a function of $I$ and $B$. Current sweeps in both bias directions are performed at each value of magnetic field by starting at $I = 0$, in order to exclude the effects of Joule heating.

We first discuss the results at zero gate voltage, $V_{g,1} = V_{g,2} = V_{g,3} = 0$ V. In Fig. 3a, we show the differential resistance map as a function of $I$ and $B$. The asymmetry between positive and negative bias is clearly visible as a tilt in the Fraunhofer-like superconducting interference lobes with respect to the current axis. The interference pattern is antisymmetric with respect to both magnetic field and current bias, a clear consequence of broken inversion and time reversal symmetry. In Fig. 3b and c we show line cuts from Fig. 3a at $B = -0.04$ mT and $B = 0.04$ mT, respectively, for positive bias direction (shown by red circles) and negative bias direction (shown by black squares). It can be clearly seen that the polarity of $\delta I_c$ is inverted when the direction of $B$ is

flipped. From the resistance map in Fig. 3a, $\delta I_c$ is extracted, and a diode efficiency $Q$ can be calculated with $Q = 2\delta I_c/(I_c^+ + |I_c^-|)$[13,47]. The experimental diode efficiency as a function $B$ is shown by the red circles in Fig. 3d. $Q$ shows periodic oscillations in $B$, displaying that the diode efficiency is tunable by a small magnetic field. In Device 1, $Q$ reaches a maximum of ~48%, comparable to or higher than previously reported values[4,13]. $Q$ oscillates but does not decrease in magnitude on application of field. These oscillations roughly follow a $\Phi_0/2$ periodicity, at consequence of the effective Josephson C$\varphi$R being $\pi$-periodic. By setting $B$ to a small negative value, the diode is tuned to have a negative polarity. At this $B$, we demonstrate supercurrent rectification of a 0.1 Hz square-wave current signal (shown in yellow in Fig. 3e), with an amplitude of $A = 0.57\,\mu A$. This amplitude is set such that $I_c^+ < A < |I_c^-|$. The device remains superconducting ($V \approx 0$ V) for the negative cycle of the square wave and has a finite voltage ($V \approx 75\,\mu V$) drop for the positive cycle. Minor punch-through errors in rectification are observed, but are believed to be due to fluctuations in the applied external field (see Supplementary Information section I and Fig. 1). We also show that the polarity of this diode can be flipped by changing the source-drain configuration to terminals 2 and 0, as discussed in the previous section (Supplementary Fig. 2).

A second device, Device 2, which is lithographically identical to Device 1, is also measured, with source terminal 2 and drain terminal 0. Device 2 shows an even higher peak efficiency of $Q_{max} \approx 68\%$. Rectification with positive polarity ($I_c^+ > |I_c^-|$) is shown on this device and only a single punch-through error is observed over 5000 cycles (Supplementary Fig. 3d). For the network model, the maximum efficiency of the diode is 55%. These devices show higher or close to this theoretical maximum number. It is important to note that the diode effect in three-terminal JJs is independent of the material platform used, being a consequence of the unconventional Josephson C$\varphi$R that arises in such devices. These results show a robust way to realize a superconducting analogue of a semiconducting diode.

In the previous section, 3TJJs were modeled as a network of three JJs with a flux through the enclosed area to illustrate that multi-terminal Josephson devices can realize this effect. However, that model did not include flux penetrating the junctions themselves or account for the possible spatially-nonuniform supercurrent density in the junction region[48–50]. To account for these effects, we replace $I_c \sin(\varphi)$-like terms in Eq. (3) by $\int J_c(x) \sin(\varphi(x)) dx$. The critical current in both bias directions can then be calculated numerically for a given $B$. To model the zero-gate-voltage case, we use a uniform current distribution, $J_c(x) = $ const in all three legs of the junction. The model derivation and parameters can be found in Supplementary Information section IV. The diode efficiency factor obtained from this updated model as a function of $B$ shows good agreement with our experimental results (Fig. 3d). The effective areas of the junction legs used in the simulation are much higher than the lithographic dimensions (see Supplementary Table I and Fig. 4 for the parameters used). This can be attributed to flux focusing effects[51,52]. As $B$ increases, flux focusing becomes weaker, resulting in a progressively increasing period of oscillation vs. $B$ in the data. This can account for small deviations between data and model observed at larger fields.

## Gate tunable Josephson diode

Applying a symmetric negative gate voltage $V_g$ to all three gates has a strong impact on the observed $I$ vs $B$ superconducting interference pattern compared to the ungated case. Comparing the data obtained at $V_g = 0$ (Fig. 3a) to that taken at $V_g = -1.5$ V (Fig. 4a), we observe that the Fraunhofer-like modulation of the lobes away from $B = 0$ is suppressed, with the lobe amplitudes becoming more uniform. Further gating to $V_g = -2.5$ V and $V_g = -4.1$ V (Fig. 4c, e) continues this trend, with the interference pattern acquiring a close resemblance to that seen in a planar SQUID. These changes in the interference pattern also have a strong effect on diode efficiency. At $B = 0.04$ mT, the diode has a

positive polarity for $V_g = 0$ V and −1.5 V, but a negative polarity for $V_g = -2.5$ V and −4.1 V. In other words, in Fig. 3c ($V_g = 0$ V) and Fig. 4b ($V_g = -1.5$ V), the critical current in the positive bias direction, $I_c^+$, is higher than the critical current in negative bias direction, $I_c^-$; however this polarity is reversed upon further application of gate voltage as seen in Fig. 4d, f where $I_c^- > I_c^+$. This polarity flip upon gating is equivalent to changing the chirality of the device. We also show diode efficiency as a function of magnetic field for all three gate voltages in Supplementary Fig. 5a, further demonstrating that for a given field, the diode efficiency and polarity can be tuned by electrostatic gating. To our knowledge, this is the first demonstration of complete gate tunablity of a Josephson diode's efficiency and polarity by changing supercurrent distribution, providing further control of this effect. This transformation of interference pattern and change of diode efficiency can be accounted for by considering the modification of $J_c(x)$ and the net flux penetrating the central region due to gating.

A planar JJ can act as a SQUID if its supercurrent density acquires two peaks close to the junction edges[48–50]. For each junction leg in our device, a negative gate voltage can act to concentrate $J_c(x)$ near the mesa edges, where the gating efficiency is weaker. This is due to formation of an electron accumulation layer at the mesa edge, which responds more weakly to gating than the middle of the leg, a consequence of the band bending effect at InAs boundaries[53–55]. The other peak in the $J_c(x)$ forms near the common central region, as the gates are designed to not cover the junction legs entirely. This results in a $J_c(x)$ resembling that of a planar SQUID in each leg. Sweeping the gate voltage to more negative values simultaneously enhances the overall contribution of these peaks to the supercurrent relative to the middle of the leg, and also introduces an asymmetry in the relative contributions of each peak (which lifts the nodes of the SQUID interference pattern). This accounts for the observed interference patterns seen in Fig. 4a, b, c.

In our simulations, we model the peaks in the supercurrent density by Lorentzian distributions. Simulated values of $Q$ using $J_c(x)$ peaking at the leg edges and in the central region (see Supplementary section IV and Fig. 4) show qualitative agreement with the experimental $Q$ values observed upon gating, as seen in Supplementary Fig. 5a,b. In this basic model, reproducing the diode polarity change requires a modification in flux in the central region. Possible reasons for this may include small fields generated due to currents in the legs near the small central region or limitations of the model when the width of the Lorentzian peaks is comparable to the size of the central area, but more work is needed to fully elucidate the origin of the observed gate-tunablity. The tuned model parameters are listed in Supplementary Table 1.

## Three-terminal diode

So far, we have demonstrated that a three-terminal Josephson device exhibits the superconducting diode effect and that this effect is a result of having more than two terminals. In the previous sections, only two-terminal measurements were performed. In those measurements terminal 2 is electrically floating. We now show that if all three terminals are biased and read out, new functionalities beyond those of two-terminal devices are possible.

We demonstrate this on Device 3, lithographically identical to Device 1. We apply a square wave current signal of amplitude $A = 0.2\,\mu A$ on terminal 1 ($I_1$) and a linearly increasing current signal on terminal 2 ($I_2$). For small values of $I_2$, $V_1 \approx 0$, but as $I_2$ is increased beyond approximately 0.3 $\mu A$, we observe rectified pulses of finite $V_1$. The magnitude of $V_1$ increases nonlinearly as a functions of $I_2$, as shown in Fig. 5a. Such non-linear intermodulation of signals resembles, for example, the activation of a neuron and may be relevant for neuromorphic computing, a promising hardware approach to artificial intelligence. The experimental result can be understood by considering the simplified network model for the device shown in Fig. 2a. For small values of $I_2$, the junction between terminals 2 and 0 remains

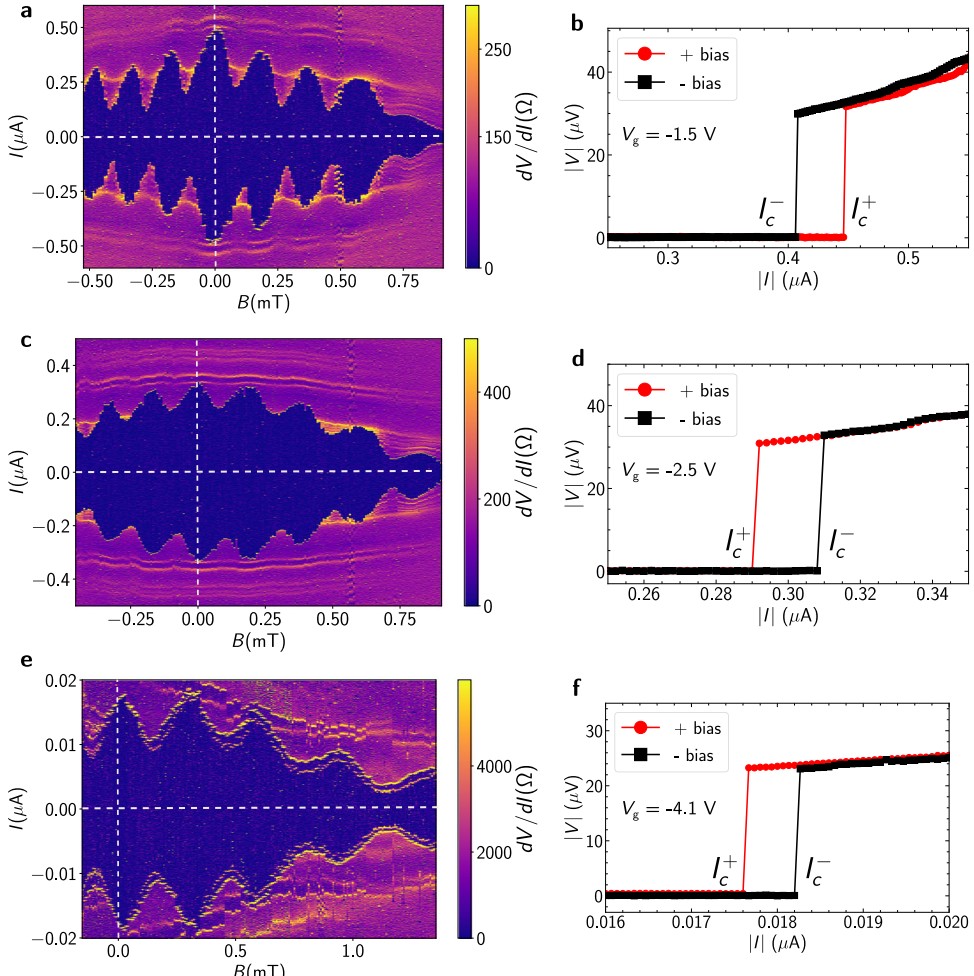

**Fig. 4 | Polarity reversal of the diode effect.** Differential resistance map of Device 1 as a function of $I$ and $B$, with all three split gates set to a value of (**a**) −1.5 V, (**c**) −2.5 V, and (**e**) −4.1 V. The crosshairs shown by white dashed lines are at zero values of $B$ and $I$. $I$–$V$ characteristics for both bias directions at $B$ = 0.04 mT, with all three split gates set to a value of (**b**) −1.5 V, (**d**) −2.5 V, and (**f**) −4.1 V, showing the reversal of the diode polarity.

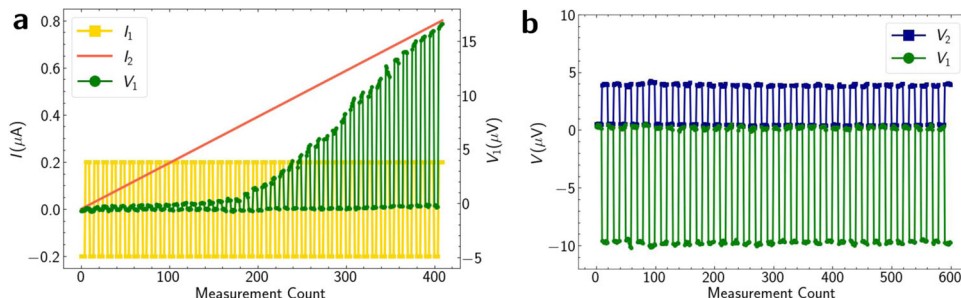

**Fig. 5 | Multi-terminal signal intermodulation and rectification. a** Measured $V_1$ (green), showing nonlinearly increasing signal amplitude demonstrating nonlinear intermodulation of applied signals $I_1$ (yellow) and $I_2$ (red). **b** Measured $V_1$ (green) with positive polarity and $V_2$ (blue) with negative polarity, demonstrating simultaneous rectification of two applied out-of-phase square wave signals on terminal 1 and 2 (not shown).

superconducting, hence $I_2$ is shunted to the ground by this Josephson junction. Since the magnitude of the square wave input to terminal 1 is less than $|I_c^{01}|$, the device output, $V_1$, is essentially zero. As $I_2$ increases above $I_c^{02}$, the net current across the junction between terminal 1 and 0 exceeds $I_c^{01}$ for the positive cycling of $I_1$, resulting in $V_1 > 0$. For negative $I_1$, the net current remains below $I_c^{01}$ and $V_1 \approx 0$.

As discussed in the previous sections, the polarity of the diode depends upon the source-drain terminals, which relate to the effective $C\varphi R$. We exploit this to show simultaneous rectification of two signals with opposite polarities. We apply two square wave signals of the same amplitude ($A = 0.6\,\mu A$) simultaneously on terminals 1 and 2, such that they are out of phase with each other. Under these conditions, we observe two simultaneously rectified signals, one on terminal 1 ($V_1$) and one on terminal 2 ($V_2$), with opposite polarities and different magnitudes, as shown in Fig. 5b.

Such device functionalities are not possible in a usual two-terminal Josephson device. This makes MTJJs uniquely suitable for applications involving multi-signal integration and rectification.

## Discussion

Our results establish a compact, yet scalable and robust approach to realize the Josephson diode effect at practically zero field (smaller in magnitude than Earth's natural field) and achieve a gate-tunable Josephson device with an unconventional C$\varphi$R. In this work, we have shown data on four devices (See Supplementary Fig. 7 for data from device 4), displaying high reproducibility of the effect. All devices studied showed the diode effect. Importantly, this multi-terminal Josephson diode effect depends only on the three-terminal nature of the device and could thus be integrated with other materials platforms, including those typically used in superconducting qubits. Multi-terminal Josephson devices made from materials with conventional C$\varphi$R could more broadly realize unconventional physics, such as semi-classical topologically-protected states[56,57]. The devices presented here could serve to realize such states. Although this requires substantial further exploration beyond the scope of this work, this points to the versatility of this approach.

We confirm that the device can be driven into an effective two-terminal regime by selectively gating two of the junction legs, where the diode effect is indeed absent (see Supplementary Fig. 3b). Inversion symmetry breaking, which is necessary for realizing non-reciprocal critical currents, is achieved by the presence of the third terminal. We emphasize that this mechanism is entirely distinct from previous works, which either relied on specific material properties, or on an imbalance of self-inductances in devices requiring large critical currents. In contrast, we demonstrate gate-tuning of the diode efficiency (magnitude and sign) by modifying $J_c(x)$ and demonstrate nonlinear signal integration enabled by the multi-terminal nature of our device, which may have applications in neuromorphic computing architectures. Previously minimal gate tunablity of superconducting diode has been demonstrated in trilayer graphene devices[8] through magnetic field training combined with simultaneous gating. The underlying mechanism for the diode effect remains an open question in that system. In contrast, we demonstrate all-electrostatic tuning of the diode efficiency, $Q$, by simply modifying $J_c(x)$.

We point out that $Q$ can be further increased by the addition of more terminals, which would lead to the generation of higher harmonics in the C$\varphi$R (generalizing Eq. (1)). If, in addition, the junctions themselves use materials that can intrinsically realize higher harmonics, this may yet further enhance $Q$[26].

Our simulations also provide a tool to model current distribution in three-terminal Josephson devices, which may help in optimizing the design for multi-terminal Josephson junctions where multiple superconducting electrodes are coupled via a central common region. We have also studied the temperature dependence of the diode efficiency, which reveals that it is robust against thermal effects at low enough temperatures where the critical current saturates (Supplementary Fig. 6). Additional integration of ferromagnetic materials may allow the device to function at precisely zero applied field, though the current devices already require a field of only a few $\mu$T.

## Methods

### Materials

The heterostructure used to fabricate the devices is grown on semi-Insulating InP(001) substrate by molecular beam epitaxy. From the bottom, the heterostructure consists of a graded buffer of $In_xAl_{1-x}As$ with $x$ ranging from 0.52 to 0.75, 25 nm $In_{0.75}Ga_{0.25}As$ superlattice. The 4.52 nm InAs quantum well is protected by a top and bottom 10.72 nm $In_{0.75}Ga_{0.25}As$ barrier. The InAs quantum well is proximitized by an epitaxial Al layer of thickness 10 nm. A 3-D schematic is shown in Fig. 1b.

### Device fabrication

Standard electron beam lithography (EBL) and wet etching techniques were used to fabricate a mesa and the Y-shaped junction area. Approximately 40 nm of $Al_2O_3$ dielectric was deposited using thermal

atomic layer deposition (ALD) at 200° C. Using EBL, split gates are defined over the junction area and gate electrodes are deposited using electron-beam evaporation of Ti/Au (5 nm/25 nm). In a separate lithography step thicker gold contacts (Ti/Au, 5 nm/200 nm) are made to the gate electrodes[58].

### Measurement details

Differential resistance on the device was obtained by low-noise DC transport measurements in a $^3$He/$^4$He dilution refrigerator. Low-pass Gaussian filtering was used to smooth numerical derivatives. Joule heating is excluded by starting the bias sweeps at zero current for both positive and negative sweep directions. A magnetic field offset $B_{offset}$ is subtracted from all the magnetic field sweeps to account for small field offsets due to residual flux in the coils, as typical for superconducting magnets. $B_{offset}$ is set by the field value where $I_c^+ = |I_c^-|$. The current square wave for the rectification demonstration is generated using a function generator and applied using a DC/AC adder box with zero DC component. The voltage drop across the device is measured using a DC voltmeter with measurement frequency higher than the applied square wave frequency.

## Data availability

Source data for the figures presented in this paper are available at the following Zenodo database [https://zenodo.org/record/7879763].

## Code availability

The data plotting code and code for the performed simulations are available at the following Zenodo database [https://zenodo.org/record/7879763].

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

## Acknowledgements
The authors acknowledge useful discussions at the virtual conference "Superconducting diode effects" organized by Virtual Science Forum. We also acknowledge L. Shani and C.J. Riggert for useful comments. All aspects of the work at UMN (except nanofabrication) and at UCSB were supported by the Department of Energy under Award No. DE-SC0019274. The development of the epitaxial growth process was supported by Microsoft Research. The device nanofabrication at UMN was supported by the National Science Foundation, under Award Number DMR-1554609. Portions of this work were conducted in the Minnesota Nano Center, which is supported by the National Science Foundation through the National Nano Coordinated Infrastructure Network (NNCI) under Award Number ECCS-1542202.

## Author contributions
M.G. and V.S.P. designed the experiments. M.G. and G.G fabricated the devices. M.G. performed the transport measurements and analyzed the data on devices at UMN under the supervision of V.S.P. M.P., J.D., and C.D. grew the heterostructure under the supervision of C. J. P. M.G., G.G., and V.S.P. co-wrote the manuscript. All authors discussed the results and provided comments on the manuscript.

## Competing interests

The authors declare no competing interests.
