## [Peer Review File · Nature Communications]

REVIEWERS' COMMENTS

Reviewer #1 (Remarks to the Author):

The authors satisfactorily addressed my comments and questions. I have one minor comment left, reported below. Therefore, provided that the other Referees give their approval, I think the paper can be published in Nature Communications.

Comment:

-The authors added in Eq. 3 (argument of the second sine) the (important) term $2\pi*n/2=\pi*n$. For consistency and to avoid confusion, I would add a $2*\pi*n$ term also in the argument of the second sine in Eq. 2 (which is numerically irrelevant, but it makes formally more clear the appearance of $2\pi*n/2$ in Eq. 3).

Reviewer #2 (Remarks to the Author):

The authors have addressed my principle concern regarding the current-phase relation.

I still think the use of the phrase "terminal" for third island is confusing and unnecessary when the device is operated in a two-terminal configuration for the diode effect. When it is actually used as an active terminal it could be referred to as such. In any case, although I do not personally insist that they change it, I think that they will find the paper less easily digested by the community.

This aspect is related to a point made by reviewer 4, that there is no fundamental difference (as far as I can tell) between SQUIDs with asymmetric inductance and SQUIDs with different numbers of junctions in each arm. One need not appeal to the magnetic field generated by the inductors that penetrate the loop -- it is enough to know that some phase drops on the inductors in same the fashion as how some phase drops on the "extra" junction of the SQUID. Thus, I agree it would be appropriate to refer to that literature more fairly in relation to their device, and emphasize instead the different aspects of their device (gate-tunability + the actual multi-terminal functionality added after the first version) as the novelty presented in their manuscript.

Reviewer #4 (Remarks to the Author):

The paper “Superconducting diode effect in a three-terminal Josephson device” is devoted, according to the authors, to “observation of the Josephson diode effect in a three-terminal Josephson device based upon an InAs quantum well two-dimensional electron gas proximitized by an epitaxial aluminum superconducting layer.” The authors claim that they “show that the Josephson diode effect in these devices is a consequence of the artificial realization of a current-phase relation that contains higher harmonics” and “show that the diode effect is an inherent property of multi-terminal Josephson devices, establishing an immediately scalable approach by which potential applications of the Josephson diode effect can be realized, agnostic to the underlying material platform,” and that “these Josephson devices may serve as gate tunable building blocks in designing topologically protected qubits.”

With respect to the previous submission to Nature Nanotechnology, the manuscript has not noticeably changed. In my opinion, the authors observed a well-known phenomenon but using slightly unusual Josephson junctions. However, these junctions have no practical applications and all three claims cited above are not supported by the data in the paper, especially the claim about building blocks in designing topologically protected qubits.

The effects described in the paper, a plus/minus asymmetry of the critical current, I_c , of a three-junction device, have nothing to do with nanotechnology or communications and are not new. They can be observed, and have been observed many times, in almost any combination of multiple Josephson junctions and inductors, macroscopic or microscopic. Moreover, a very special care needs to be taken to design and make a multi-terminal “Josephson device” which would not show an asymmetry of the Josephson critical current. For instance, the entire field of superconductor digital electronics, with tens of thousands of publications, is based on the fact that a network of Josephson junctions and inductors has a critical current which depends on the location of the current terminals, currents and magnetic fluxes applied to the network. This allows to switch the required Josephson junctions in the network and perform logic operations. All these networks have no inversion symmetry a hence have different critical currents in the up and down directions.

The theoretical treatment of the three-junction devices presented in the paper, equations (1)-(3), is also not quite correct.

These equations do not take into account a finite inductance between the pairs of junctions and the phase drops associated with the presence of these inductors, i.e., the loop inductance. Without this inductance, an external magnetic flux ϕ_e cannot be applied to the three-junction loop. I.e., a magnetic flux cannot couple to a loop without inductance. Hence, equations (1)-(3) cannot be written in this simple form and need to be modified. In general, any asymmetry between the critical currents of the junctions or inductances will lead to the device critical current asymmetry. This is a trivial and well-known effect.

Also, the initial claim “that the Josephson diode effect in these devices is a consequence of the artificial realization of a current-phase relation that contains higher harmonics” is not correct. First of all, it is not proper to apply the term “current-phase” relationship to a circuit; see below. Secondly, the presence of higher harmonics is not important for the I_c asymmetry. The important is only the presence of a phase shift.

Josephson device is usually a synonym to Josephson junction or a Josephson diode – a two-terminal device. Diode, by definition, is an electronic component with two electrodes. The name itself comes from the Greek “di” (meaning two) + ode, a shortened form of electrode. Even a two-junction device is not called a Josephson device; it is called SQUID, although it still has only two terminals. Observation of electric current flow asymmetry (rectification) in the two-terminal devices requires an inversion symmetry breaking in the diode by either an internal (or external) electric field or by an internal or external magnetic field, or by spin-orbit interactions in the device, etc. These are fundamental physics effects. However, any three-terminal device is not a diode. It is a triode and the mere presence of the third electrode breaks the symmetry between the other two. This is a trivial effect, not worth publishing in Nature.

The authors write on p. 2: “asymmetric superconducting quantum interference devices (SQUIDs) can also realize non-reciprocal transport [19–24] by using large and imbalanced self-inductances,

typically in all-metallic Josephson devices, which has precluded electrostatic gating. In a recent work, it was also theorized that interferometers based upon higher-harmonic Josephson junctions can realize the Josephson diode effect, with implementation relying on highly transparent quantum point contact JJs [25]. These approaches are material specific and difficult to scale for potential applications of the Josephson diode effect, such as dissipationless electronics.”

These statements are not quite correct. These asymmetrical SQUIDs have been used for many years as rectifying diodes in ac-to-dc converters in many superconductor integrated circuits and can be easily scaled up; see e.g., ref. [24] in the paper and V. K. Semenov, E. B. Golden and S. K. Tolpygo, "SFQ bias for SFQ digital circuits," IEEE Trans. Appl. Supercond., vol. 31, no. 5, pp. 1-7, Aug. 2021, Art no. 1302207, doi: 10.1109/TASC.2021.3067231. Also, there is no “dissipationless electronics.” It does not exist and is not possible.

Please note, in all the referenced above cases, the devices are two-terminal devices and thus are real diodes. The inductors do not need to be “large and imbalanced self-inductances,” they can be small and kinetic, the imbalance can be created by the Josephson junctions, e.g., by placing one junction in one arm and two junctions in series in the other arm, or by the difference in their critical currents, by a mutual inductance between the SQUID arms, or by the self-induced magnetic flux, or by a magnet placed inside, or by anything. The junctions do not have to be “all-metallic.” The only requirement for

rectification (“Josephson diode effect”) is an asymmetry of the two-terminal SQUID with respect to the current feeding points.

The three- and more terminals “Josephson devices” are not different from the asymmetric SQUIDs discussed above and will almost always show asymmetry of the critical currents. Inductance of the third junction (the phase drop across it, which is the same) is what creates this asymmetry. Therefore, the asymmetrical critical current observed by the authors in their Al-proximitized-2D-electron-gas Josephson junctions device is a trial effect of the three connected junctions and nonzero parasitic inductance between them. It can be observed, and has been observed many times, in any superconductor integrated circuit. A 2D electron gas, quantum wells, and millikelvin temperature are not needed for this.

The authors use a concept of “a current-phase relationship” for their multiterminal, multijunction device. The current-phase relationship has meaning for a single Josephson junction, where it relates the current through the junction to the phase difference between its electrodes. It is a fundamental property of a junction. However, in a network of junctions and inductors, the current between any two terminals is a very complicated function of the phase differences across other junctions. It depends on the position of the terminals, on the presence of inductors and magnetic fluxes in the network, π -junctions, $\pi/2$ junctions, etc. It is no longer a universal property. In almost all cases this dependence is not tractable analytically and specialized superconductor circuit simulators such as PSCAN and PSCAN2, JoSim, Spectre, WRSPICE, etc. are used to solve nonlinear Kirchhoff equations and find critical currents in these networks and determine their asymmetry. Therefore, a concept of a current-phase relationship for a complex circuit is meaningless from my point of view.

The electrostatic gating described by the authors has no practical applications in superconductor electronics or for qubits because it requires too large voltages to produce very small effects. That is, these “superconducting field-effect transistors” have no gain; and no gain means no use.

The manuscript has parts, especially in the Introduction and Abstract, which are not relevant to the content of the paper. For instance, “Experimental investigations of Andreev bound state spectra in multi-terminal Josephson devices have attracted considerable attention recently due to proposed topologically protected subgap states [26–32]. Despite technical challenges in realizing these subgap states, other interesting transport phenomena such as multi-terminal Andreev reflections [33–35], fractional Shapiro steps [36, 37], correlated phase dynamics [38, 39], and semiclassical topological states [40] have been demonstrated. The discovery and characterization of nonreciprocal supercurrent flow in these devices stands to impact and expand upon these phenomena.” The subject of the paper has nothing to do with all these topics and references, and the results have no relevance to them.

I believe that the experimental data presented in the paper are (most likely) correct. However, they do not present any new or important effects and do not create any substantial new knowledge which would warrant publication in Nature Communications. After a revision and improving the theoretical treatment, the paper may become suitable for Supercond. Sci. Technol. or IEEE TAS.

Re: NCOMMS-23-12189-T Superconducting Diode Effect in a Three-terminal Josephson Device by Mohit Gupta, Gino V. Graziano, Mihir Pendharkar, Jason T. Dong, Connor P. Dempsey, Chris Palmstrøm, Vlad S. Pribiag

We thank the Reviewers for reviewing our manuscript and providing valuable feedback. We believe that the following responses adequately address all the remaining comments.

We give below detailed responses to each point made by the Reviewers. The remarks of each Reviewer are quoted verbatim and shown in **bold face font**; our responses are in normal text. The main revisions to the manuscript are described below. In addition, the main revisions are indicated in blue in the revised manuscript.

Report of Reviewer #1 –NCOMMS-23-12189-T

The authors satisfactorily addressed my comments and questions. I have one minor comment left, reported below. Therefore, provided that the other Referees give their approval, I think the paper can be published in Nature Communications.

We are happy to see that the Reviewer thinks we have addressed all their questions satisfactorily. We thank them for recommending publication in Nature Communications.

Comment: -The authors added in Eq. 3 (argument of the second sine) the (important) term $2\pi*n/2=\pi*n$. For consistency and to avoid confusion, I would add a $2*\pi*n$ term also in the argument of the second sine in Eq. 2 (which is numerically irrelevant, but it makes formally more clear the appearance of $2\pi*n/2$ in Eq. 3).

We have incorporated this edit in the revised manuscript.

Report of Reviewer #2 –NCOMMS-23-12189-T

The authors have addressed my principle concern regarding the current-phase relation.

I still think the use of the phrase "terminal" for third island is confusing and unnecessary when the device is operated in a two-terminal configuration for the diode effect. When it is actually used as an active terminal it could be referred to as such. In any case, although I do not personally insist that they change it, I think that they will find the paper less easily digested by the community.

We are happy to note that Reviewer's concerns regarding the current-phase relation were addressed. We believe that the distinction between two and three terminal measurements was made

sufficiently clear in our previous version. We have made minor edits in our revised manuscript to clarify this point further. We thank the Reviewer for their helpful suggestion.

This aspect is related to a point made by reviewer 4, that there is no fundamental difference (as far as I can tell) between SQUIDs with asymmetric inductance and SQUIDs with different numbers of junctions in each arm. One need not appeal to the magnetic field generated by the inductors that penetrate the loop – it is enough to know that some phase drops on the inductors in same the fashion as how some phase drops on the "extra" junction of the SQUID. Thus, I agree it would be appropriate to refer to that literature more fairly in relation to their device, and emphasize instead the different aspects of their device (gate-tunability + the actual multi-terminal functionality added after the first version) as the novelty presented in their manuscript.

We thank the Reviewer for their comment. The equations presented in our work already account for the phase drop across the different junctions in the network, leading to an effective current-phase relation with higher harmonics. We have now cited all the relevant works recommended by Reviewer #4. We believe no additional discussion is necessary as the operating principle of the device is fundamentally different.

Report of Reviewer #4 –NCOMMS-23-12189-T

The paper “Superconducting diode effect in a three-terminal Josephson device” is devoted, according to the authors, to “observation of the Josephson diode effect in a three-terminal Josephson device based upon an InAs quantum well two-dimensional electron gas proximitized by an epitaxial aluminum superconducting layer.” The authors claim that they “show that the Josephson diode effect in these devices is a consequence of the artificial realization of a current-phase relation that contains higher harmonics” and “show that the diode effect is an inherent property of multi-terminal Josephson devices, establishing an immediately scalable approach by which potential applications of the Josephson diode effect can be realized, agnostic to the underlying material platform,” and that “these Josephson devices may serve as gate tunable building blocks in designing topologically protected qubits.”

With respect to the previous submission to Nature Nanotechnology, the manuscript has not noticeably changed. In my opinion, the authors observed a well-known phenomenon but using slightly unusual Josephson junctions. However, these junctions have no practical applications and all three claims cited above are not supported by the data in the paper, especially the claim about building blocks in designing topologically protected qubits.

The effects described in the paper, a plus/minus asymmetry of the critical current, I_c , of a three-junction device, have nothing to do with nanotechnology or

communications and are not new. They can be observed, and have been observed many times, in almost any combination of multiple Josephson junctions and inductors, macroscopic or microscopic. Moreover, a very special care needs to be taken to design and make a multi-terminal “Josephson device” which would not show an asymmetry of the Josephson critical current. For instance, the entire field of superconductor digital electronics, with tens of thousands of publications, is based on the fact that a network of Josephson junctions and inductors has a critical current which depends on the location of the current terminals, currents and magnetic fluxes applied to the network. This allows to switch the required Josephson junctions in the network and perform logic operations. All these networks have no inversion symmetry a hence have different critical currents in the up and down directions.

The theoretical treatment of the three-junction devices presented in the paper, equations (1)-(3), is also not quite correct. These equations do not take into account a finite inductance between the pairs of junctions and the phase drops associated with the presence of these inductors, i.e., the loop inductance. Without this inductance, an external magnetic flux ϕ_e cannot be applied to the three-junction loop. I.e., a magnetic flux cannot couple to a loop without inductance. Hence, equations (1)-(3) cannot be written in this simple form and need to be modified. In general, any asymmetry between the critical currents of the junctions or inductances will lead to the device critical current asymmetry. This is a trivial and well-known effect.

Also, the initial claim “that the Josephson diode effect in these devices is a consequence of the artificial realization of a current-phase relation that contains higher harmonics” is not correct. First of all, it is not proper to apply the term “current-phase” relationship to a circuit; see below. Secondly, the presence of higher harmonics is not important for the I_c asymmetry. The important is only the presence of a phase shift.

Josephson device is usually a synonym to Josephson junction or a Josephson diode – a two-terminal device. Diode, by definition, is an electronic component with two electrodes. The name itself comes from the Greek “di” (meaning two) + ode, a shortened form of electrode. Even a two-junction device is not called a Josephson device; it is called SQUID, although it still has only two terminals. Observation of electric current flow asymmetry (rectification) in the two-terminal devices requires an inversion symmetry breaking in the diode by either an internal (or external) electric field or by an internal or external magnetic field, or by spin-orbit interactions in the device, etc. These are fundamental physics effects. However, any three-terminal device is not a diode. It is a triode and the mere presence of the third electrode breaks the symmetry between the other two. This a trivial effect, not worth publishing in Nature.

The authors write on p. 2: “asymmetric superconducting quantum interference devices (SQUIDs) can also realize non-reciprocal transport [19–24] by using large and imbalanced self-inductances, typically in all-metallic Josephson devices, which

has precluded electrostatic gating. In a recent work, it was also theorized that interferometers based upon higher-harmonic Josephson junctions can realize the Josephson diode effect, with implementation relying on highly transparent quantum point contact JJs [25]. These approaches are material specific and difficult to scale for potential applications of the Josephson diode effect, such as dissipationless electronics.” These statements are not quite correct. These asymmetrical SQUIDS have been used for many years as rectifying diodes in ac-to-dc converters in many superconductor integrated circuits and can be easily scaled up; see e.g., ref. [24] in the paper and V. K. Semenov, E. B. Golden and S. K. Tolpygo, ”SFQ bias for SFQ digital circuits,” IEEE Trans. Appl. Supercond., vol. 31, no. 5, pp. 1-7, Aug. 2021, Art no. 1302207, doi: 10.1109/TASC.2021.3067231. Also, there is no “dissipationless electronics.” It does not exist and is not possible.

Please note, in all the referenced above cases, the devices are two-terminal devices and thus are real diodes. The inductors do not need to be “large and imbalanced self-inductances,” they can be small and kinetic, the imbalance can be created by the Josephson junctions, e.g., by placing one junction in one arm and two junctions in series in the other arm, or by the difference in their critical currents, by a mutual inductance between the SQUID arms, or by the self-induced magnetic flux, or by a magnet placed inside, or by anything. The junctions do not have to be “all-metallic.” The only requirement for rectification (“Josephson diode effect”) is an asymmetry of the two-terminal SQUID with respect to the current feeding points.

The three- and more terminals “Josephson devices” are not different from the asymmetric SQUIDS discussed above and will almost always show asymmetry of the critical currents. Inductance of the third junction (the phase drop across it, which is the same) is what creates this asymmetry. Therefore, the asymmetrical critical current observed by the authors in their Al-proximitized-2D-electron-gas Josephson junctions device is a trial effect of the three connected junctions and nonzero parasitic inductance between them. It can be observed, and has been observed many times, in any superconductor integrated circuit. A 2D electron gas, quantum wells, and millikelvin temperature are not needed for this.

The authors use a concept of “a current-phase relationship” for their multi-terminal, multijunction device. The current-phase relationship has meaning for a single Josephson junction, where it relates the current through the junction to the phase difference between its electrodes. It is a fundamental property of a junction. However, in a network of junctions and inductors, the current between any two terminals is a very complicated function of the phase differences across other junctions. It depends on the position of the terminals, on the presence of inductors and magnetic fluxes in the network, pi-junctions, pi/2 junctions, etc. It is no longer a universal property. In almost all cases this dependence is not tractable analytically and specialized superconductor circuit simulators such as PSCAN and PSCAN2, JoSim, Spectre, WRSPICE, etc. are used to solve nonlinear Kirchhoff equations

and find critical currents in these networks and determine their asymmetry. Therefore, a concept of a current-phase relationship for a complex circuit is meaningless from my point of view.

The electrostatic gating described by the authors has no practical applications in superconductor electronics or for qubits because it requires too large voltages to produce very small effects. That is, these “superconducting field-effect transistors” have no gain; and no gain means no use.

The manuscript has parts, especially in the Introduction and Abstract, which are not relevant to the content of the paper. For instance, “Experimental investigations of Andreev bound state spectra in multi-terminal Josephson devices have attracted considerable attention recently due to proposed topologically protected subgap states [26–32]. Despite technical challenges in realizing these subgap states, other interesting transport phenomena such as multi-terminal Andreev reflections [33–35], fractional Shapiro steps [36, 37], correlated phase dynamics [38, 39], and semiclassical topological states [40] have been demonstrated. The discovery and characterization of nonreciprocal supercurrent flow in these devices stands to impact and expand upon these phenomena.” The subject of the paper has nothing to do with all these topics and references, and the results have no relevance to them.

I believe that the experimental data presented in the paper are (most likely) correct. However, they do not present any new or important effects and do not create any substantial new knowledge which would warrant publication in *Nature Communications*. After a revision and improving the theoretical treatment, the paper may become suitable for *Supercond. Sci. Technol.* or *IEEE TAS*.

We thank the Reviewer for reviewing our manuscript and providing their questions and comments. Indeed, we agree with the Reviewer that our device has little to do with “communications”. The claims regarding innovation in any work can be subjective. However, we would like to reiterate the primary innovations of our work:

- We demonstrate that a three-terminal Josephson device generates an unconventional CPR, which along with a small applied magnetic field leads to the Josephson diode effect.
- Using a gate-tunable material, we show full control of the Josephson diode polarity and continuous tuning of the efficiency by electrostatic control of $J_c(x)$. To our knowledge, this is the first demonstration of a superconducting diode that can be tuned solely by gating.
- Finally, the multi-terminal nature of our approach enables further functionalities. Specifically, in the revised manuscript we demonstrated non-linear signal intermodulation and simultaneous rectification of multiple pulses with different polarities.

We would like to note that in the last round we provided a detailed explanation addressing the Reviewer’s technical comments about the role of imbalanced kinetic and self inductances, the difference between our work and work on asymmetric SQUIDs, and the importance of gate tunability in our work. As we showed there, those concerns were not applicable. In this

reply, Reviewer # 4 simply reiterates their comments almost verbatim and offers no counter arguments regarding our aforementioned explanations. Therefore, we believe no outstanding technical concerns remain regarding our work.